# Validation of MotionWatch8 Actigraphy Against Polysomnography in Menopausal Women Under Warm Conditions

**DOI:** 10.3390/s25103040

**Published:** 2025-05-12

**Authors:** Xinzhu Li, Mark Halaki, Chin Moi Chow

**Affiliations:** Sydney School of Health Sciences, Faculty of Medicine and Health, University of Sydney, Camperdown, NSW 2050, Australia; cynthia.li@sydney.edu.au (X.L.); chin-moi.chow@sydney.edu.au (C.M.C.)

**Keywords:** validation, PSG, polysomnography, actigraphy, sleep, menopause, MotionWatch8

## Abstract

**Highlights:**

**Abstract:**

This study evaluated the agreement between MotionWatch8 actigraphy and polysomnography (PSG) in measuring sleep parameters among menopausal women under controlled 30 °C laboratory conditions. Sixteen peri- and post-menopausal women (age: 51.4 ± 4.2 years, BMI: 26.0 ± 3.1 kg/m^2^) contributed 59 nights of simultaneous recordings, with parameters analyzed using Bland–Altman plots, linear mixed model analysis, and epoch-by-epoch comparisons. Results showed MotionWatch8 significantly overestimated total sleep time by 18.6 min and sleep efficiency by 3.5%, while underestimating sleep onset latency by 11.2 min and wake after sleep onset by 9.1 min compared to PSG. Significant proportional errors were observed, particularly for participants with prolonged sleep onset latency, high wake after sleep onset, and lower sleep efficiency. Epoch-by-epoch analysis revealed high sensitivity for sleep detection (94.8%) but low specificity for wake detection (33.1%), with 87.3% overall accuracy. These findings demonstrate that MotionWatch8 may be less reliable for individuals with more extreme sleep characteristics, such as insomnia, as measurement accuracy declines with increasing severity of sleep disturbances, highlighting the need for caution when using this device for detailed sleep assessments in clinical populations with sleep disturbances.

## 1. Introduction

Menopause is a significant life transition that can drastically affect a woman’s health, with sleep disturbances being one of the most reported issues [1,2]. Sleep disturbances affect around 40–60% of menopausal women [3,4]. Vasomotor symptoms, such as hot flashes and night sweats, reported by 75% of menopausal women, contribute to sleep disturbances [3,5,6]. Despite their high prevalence and considerable impact, sleep disturbances in menopausal women are often overlooked or trivialised. Accurately capturing sleep disturbances in menopausal women is critical to understanding their extent and developing effective approaches to address their impact.

While polysomnography (PSG), the gold standard for sleep measurement, captures important data on sleep architecture and patterns [7], it is impractical for tracking sleep over extended periods. Nevertheless, the SWAN sleep study demonstrated the feasibility of conducting three-night PSG recordings, albeit in-home studies [8]. Actigraphy, a non-invasive method that tracks movement, offers an accessible alternative in the home and field setting for long-term monitoring [9,10,11]. Since its introduction in the 1970s, actigraphy has been widely used and validated in diverse populations [9,12], consistently demonstrating high concordance with PSG results across different conditions, such as insomnia, circadian rhythm disorders, and chronic health conditions, as well as across age groups ranging from children, adolescents, to older adults [13,14,15,16,17,18,19]. A recent validation study by Waki et al. [20] compared the MotionWatch8 with a commercial wearable (Fitbit Inspire HR) in healthy adults.

However, despite its widespread application, few studies have specifically validated actigraphy in menopausal women, a population uniquely affected by sleep disturbances due to hormonal fluctuations. This gap is especially relevant given that warm environmental conditions are known to trigger vasomotor symptoms such as hot flashes and night sweats [21,22,23], which in turn exacerbate sleep disturbances in this population. Although the MotionWatch© has been validated against PSG in some conditions, including in non-shift-working adult population [15,19,24,25,26,27,28,29,30], its performance under warm environmental conditions and in hormonally sensitive populations such as menopausal women remains unclear. This issue is particularly important in regions regularly experiencing summer temperatures exceeding 30 °C, such as Sydney and Melbourne (Australia) [31,32], Cancun (Mexico) with an average temperature of 30.5 °C in January [33], many coastal cities in China [34], and the eastern United States [35]. Countries like Qatar and Kenya experience such temperatures all year round [36,37]. Previous research by Shin et al. [38] demonstrated that ambient temperature can influence the behaviour and accuracy of actigraphy devices in sleep measurements, underscoring the need for validation in controlled warm conditions. This study therefore aims to address this gap by validating the effectiveness of the MotionWatch8© (CamNtech Ltd., Cambridgeshire, UK) against PSG in measuring sleep parameters among menopausal women under controlled warm laboratory conditions.

## 2. Method

### 2.1. Participants

A total of 16 peri- or post-menopausal women from the Greater Sydney Area, NSW, Australia, participated in this study. Participant characteristics are summarised in Table 1.

**Inclusion criteria:** Participants were required to be experiencing vasomotor menopausal symptoms, defined as a vasomotor scale score ≥ 2 on the Modified Greene Climacteric Scale [39] or with self-reported daily hot flashes or night sweats. Additional criteria included irregular menstruation period due to menopause transition over the past 12 months (perimenopausal), cessation of menstruation for at least one year (postmenopausal), or bilateral oophorectomy (surgically induced menopause) at least six weeks prior to screening.

**Exclusion criteria:** Individuals were excluded if they were shift workers, had sleep difficulties or sleep disorders, chronic health conditions affecting sleep (e.g., cardiovascular diseases, pulmonary diseases, diabetes, or metabolic syndrome), were on hormone replacement therapy (HRT) or were taking medications that could impact sleep.

**Recruitment strategy:** Participants were mainly recruited via advertisements through flyers, local newspapers, social media, and referrals from a recruiting agency, Trialfacts (Available online: https://trialfacts.com (accessed on 6 February 2020)).

Actigraphy and PSG data were collected simultaneously during overnight sleep studies.

### 2.2. Study Procedures

During the laboratory visits, participants underwent screening by completing several questionnaires about health history, sleep-related scales, menopausal status, and menopause symptoms. Demographic information was collected. Physical measurements, including height, weight, waist and neck circumference, and blood pressure were recorded.

Eligible participants completed four overnight sleep studies inconsecutively with at least one night interval between studies in a controlled laboratory environment. The bedroom was maintained at a temperature of 30 °C (30.1 ± 0.5 °C) and relative humidity of 50% (46.4 ± 2.7%) as part of a broader investigation into the effects of sleepwear fibre type on sleep quality.

On each study night, participants arrived approximately 4.5–5 h before their usual bedtime. A standardised dinner was served at 4 h before their usual bedtime, following a physical examination. Participants were encouraged to maintain adequate fluid intake during the day but to restrict water consumption within two hours before bedtime to minimize nocturnal awakenings. PSG setup, including electrode and sensor placement, began 1.5–2 h before bedtime. Participants remained in the waiting area before transitioning to the sleeping room 20 min prior to lights-off. The lights-off and lights-on times corresponded to participants’ usual bedtimes and wake-up times, ensuring adherence to their typical sleep routines. During the PSG study, participants wore the MotionWatch8© on their non-dominant wrist for the entire sleep period to capture actigraphy data concurrently.

### 2.3. PSG Recording

Overnight PSG sleep data were collected using a Compumedics E-series or W-series Sleep system (Compumedics Australia Pty Ltd., Abbotsford, VIC, Australia). A standardised procedure for the placement of 10–20 electrodes was followed according to the American Academy of Sleep Medicine (AASM) guidelines [40].

Electrode placement included the following:**Electroencephalogram (EEG):** Five scalp electrodes referenced to mastoid processes (C3-M2, C4-M1, O1-M2, O2-M1, F3-M2), Cz on the scalp as reference, and one ground electrode Fpz on forehead.**Electrooculogram (EOG):** Bilateral electrodes placed near the outer canthus of each eye.**Electromyogram (EMG):** Submental electrodes placed under the chin to record muscle activity.**Electrocardiogram (ECG):** Electrodes placed on the chest to monitor heart activity.

All signals were sampled at a frequency of 256 Hz.

### 2.4. PSG Data Scoring

The PSG recordings were scored by an external independent experienced scorer according to the AASM manual [40]. A report was generated using the Compumedics Profusion PSG4 software Version 4 (Compumedics Australia Pty Ltd., Abbotsford, VIC, Australia) including the following items: report start and end time/duration, lights-off and lights-on time, time available for sleep, sleep latency, REM latency, sleep period start and end time/duration, wake after sleep onset, total sleep time (minutes), NREM sleep, sleep stages (N1/N2/N3/REM) time and proportion, and sleep efficiency. The stage of each epoch was also exported.

### 2.5. MotionWatch8 Recording and Scoring

The MotionWatch8 was selected for this study due to its widespread clinical and research use, compatibility with legacy actigraphy scoring protocols, and manufacturer-supported algorithms validated for 30 s epochs. It has been previously validated in the general adult population [41], enabling comparisons with earlier work while allowing us to explore its applicability in a thermally challenging setting. Recordings were taken in 30 s epochs using MotionWatch Mode 1, which is an epoch-based recording mode using a single axis algorithm and peak detection. Participants wore it on the non-dominant wrist.

The actigraphy recordings were processed using the software’s algorithm (MotionWare 1.2.28, CamNtech Ltd., Cologne, Germany). This algorithm, which is not publicly disclosed in detail, is designed by the manufacturer to mimic legacy actigraphy devices (e.g., Actiwatch). The movement thresholds used for sleep/wake classification are proprietary and cannot be adjusted by the user. Lights-off and lights-on times for each sleep interval were manually checked and marked by reviewing the light level, and activity counts and were cross-checked and adjusted based on the participant’s documented sleep log to improve the accuracy of the marked rest interval. The software algorithm then automatically calculated the sleep statistical data.

### 2.6. Data Analysis

To assess the agreement between PSG and actigraphy, both sleep summary statistics and epoch-by-epoch analyses were performed using data across the sleep period. Nights with incomplete actigraphy or PSG recordings were excluded from the analysis.

Bland–Altman plots were used to assess the agreements between MotionWatch8 and PSG for the sleep outcome variables of total sleep time (TST), sleep efficiency (SE), sleep onset latency (SOL) and wake after sleep onset (WASO). Each night’s data was treated as an independent observation in the analysis, meaning that data from different nights for the same participant were analysed separately, rather than averaged or combined. In the Bland–Altman plots, each data point represents the sleep parameters recorded for a single night, regardless of whether the nights belonged to the same or different participants. The differences for each sleep variable (displayed on the y-axis) were calculated as actigraphy outcome minus PSG outcome, and the mean of the two measures was displayed on the x-axis. The limits of agreements (LOA) were calculated as the mean difference ±1.96 × standard deviation, which indicated the range of values expected for 95% of individuals. To further evaluate the systematic bias and potential dependency of measurement differences on the size of the measurement, statistical significance tests were performed using SPSS. A one-sample *t*-test was used to examine whether the mean difference (offset) for each sleep variable significantly deviated from zero. Linear regression analyses were conducted to assess whether the differences (dependent variable) were significantly associated with the mean values (independent variable). The regression slope was tested for significance to determine if the measurement bias varied across the range of measurement values.

To assess the agreement between PSG and MotionWatch8, sleep parameters, including SOL, TST, SE, and WASO were compared using a linear mixed model (LMM) in SPSS. The LMM approach was chosen to account for the repeated measures design as each participant contributed up to four nights of data, and to incorporate both fixed and random effects. The model included device (PSG vs. MotionWatch8) as a fixed factor to evaluate the main effects of measurement method and visit number (night-to-night variability) as a repeated fixed factor. Study ID was included as a random intercept to account for within-subject correlation across repeated measures, with a first-order autoregressive covariance structure (AR1) specified. Estimated Marginal Means (EM Means) were generated for device to compare main effects. The model employed restricted maximum likelihood estimation (REML) for parameter estimation and used the Satterthwaite approximation for calculating degrees of freedom.

Epoch-by-epoch comparison was made to calculate sensitivity, specificity, and accuracy. Only the epochs where both the actigraphy and PSG have stages were analysed. Sleep epochs were coded as 1 and awake epochs were coded as 0. Sensitivity reflects sleep agreement and was calculated as the number of true sleep (TP)/(TP+ number of false wake epoch (FN)). Specificity measures wake agreement was calculated as the number of true wake epochs (TN)/(TN + number of false sleep epoch (FP)). Accuracy shows the overall performance of sleep and wake detection for the actigraphy against PSG, which was calculated as (TP + TN)/(TP + TN + FP + FN).

All analysis was conducted using SPSS v.29.0.1.0 and Microsoft Excel. The level of significance was set at *p* < 0.05.

## 3. Result

After excluding nights with technical issues, a total of 59 nights of valid recordings from 16 participants were included for analysis. Participants’ ages ranged from 45 to 63 years old (51.4 ± 4.2 years).

### 3.1. Bland–Altman Plots

The Bland–Altman plots for SOL, TST, SE, and WASO are displayed in Figure 1. The MotionWatch8 tended to overestimate TST by an average of 18.6 min, with LOA ranging from −56.8 min to 94.0 min. No proportional bias was found for TST (*p* = 0.819, R^2^ = 0.0009). It overestimated SE by 3.5%, with LOA between −11.7% and 18.7% and a significant proportional bias. On the other hand, the MotionWatch8 underestimated SOL by an average of 11.2 min, with LOA from −53.6 to 31.3 min. The proportional error was significant, particularly for participants with sleep onset latency greater than 30 min. Similarly, WASO was underestimated by 9.06 min, with LOA ranging from −74.80 min to 56.68 min, with a significant dispersion from the mean where WASO was greater than 55 min. Table A1 provides the mean differences, lower and upper limits of agreement (LOA), ranges, the *p* value for offset, and regression line for SOL, TST, SE, and WASO.

### 3.2. Linear Mixed Model

Table 2 summarises the linear mixed model (LMM) analysis results for sleep onset latency (SOL), wake after sleep onset (WASO), total sleep time (TST), and sleep efficiency (SE). The table displays the estimated marginal means (mean ± std. error) for each device (PSG and MotionWatch8), along with the F-values and p-values for the main effect of device.

Significant differences were observed between PSG and MotionWatch8 for SOL, TST, SE, and as indicated by *p*-values below 0.05. Additionally, a marginally significant difference was observed for WASO between the two devices (*p* = 0.063).

### 3.3. Epoch-by-Epoch Comparison

Table 3 displays the comparison results for sensitivity for sleep, specificity for wake, and accuracy between the classifications of sleep and wake.

## 4. Discussion

The present study highlights the utility and limitations of MotionWatch8 for sleep assessment in menopausal women under controlled warm laboratory conditions. Our study validated the MotionWatch8 against PSG, revealing that MotionWatch8 systematically overestimates sleep duration and underestimates wakefulness-related metrics, particularly in participants with higher SOL, WASO, and lower SE values.

Specifically, Bland–Altman plots revealed that MotionWatch8 overestimated total sleep time (TST) and sleep efficiency (SE), while underestimating sleep latency (SOL) and wake after sleep onset (WASO), with significant proportional errors observed for SE, SOL, and WASO. The linear mixed model (LMM) analysis results indicated significant differences between PSG and MotionWatch8 for TST, SE, and SL, with MotionWatch8 tending to overestimate TST and SE while underestimating SOL compared to PSG. Additionally, a trend toward significance was observed for WASO between the two devices (*p* = 0.063), with MotionWatch8 tending to underestimating WASO compared to PSG.

Significant proportional biases were observed for SOL, SE, and WASO as indicated by their regression slopes, while no proportional bias was found for TST (*p* = 0.819, R^2^ = 0.0009). This indicates that while MotionWatch8 performs satisfactorily for objective measurement of sleep parameters in individuals with regular sleep patterns, it may not be suitable for populations with disrupted sleep, e.g., insomnia. Specifically, the proportional bias for SOL (R^2^ = 0.65) and WASO (R^2^ = 0.20) suggests that device inaccuracy worsens with higher disturbance severity. These biases have clinical implications as sleep onset of >30 min and high WASO approach diagnostic thresholds for insomnia (DSM-5) [42,43]. Furthermore, an underestimation of SOL by an average of 11.2 min and WASO by 9.1 min, as observed in our study, may result in failure to meet clinical diagnostic thresholds, potentially leading to false negatives in insomnia screening or underestimation of symptom severity. These limitations stem from MotionWatch8’s tendency to misclassify periods of quiet wakefulness as sleep and its difficulty in detecting fragmented or disrupted sleep patterns.

Epoch-by-epoch comparisons demonstrated high sensitivity (94.8%) and overall moderate accuracy (87.3%), but low specificity (33.1%) for wake detection. These findings align with previous validation studies of actigraphy devices [19]. Marino et al. [15] and Lichstein et al. [28] both reported similar trends in Actiwatch devices, attributing these overestimations to the reliance on movement-based algorithms that often misclassify quiet wakefulness as sleep. Likewise, studies [27,44] highlighted actigraphy’s inherent limitations in distinguishing wake epochs during periods of minimal movement. Our study extended these findings by observing more pronounced proportional errors for SOL and WASO compared to prior research, potentially due to the controlled laboratory environment and population characteristics. For instance, Full et al. [27] found that SOL errors were less severe in healthy adult populations, potentially due to their shorter sleep latency ranges. While MotionWatch8 demonstrated high sensitivity (94.8%) for sleep detection, its low specificity (33.1%) for wake detection highlights its limitations in distinguishing wake epochs during periods of minimal movement [45]. These findings align with Kosmadopoulos et al. [44], who observed similar challenges in shift workers with irregular schedules. The lack of significant differences in WASO observed in our study differs from some previous findings, potentially due to differences in laboratory conditions or sample characteristics. Moreover, our findings suggest that MotionWatch8’s accuracy diminishes significantly in participants with extended SOL or fragmented sleep patterns, which may be a result of its algorithmic limitations under fixed laboratory conditions. These discrepancies highlight the importance of considering device-specific calibration and population-specific factors when interpreting actigraphy data.

It is worth noting that the study sample comprised predominantly Caucasian women (75%) whose menopause occurred naturally (94%). Previous research has demonstrated that both menopausal type and ethnicity can significantly affect sleep characteristics. Extreme sleep characteristics may reduce the level of concordance by actigraphy devices. For example, the sleep characteristics of our participants may show fewer extreme parameters, contrasting with women who displayed more severe sleep disturbances who had surgical menopause due to abrupt hormonal changes [4,22]. The SWAN sleep study also indicated that African-American women exhibited longer sleep onset latency and lower sleep efficiency compared to Caucasian women [8]. Notably, there is no evidence to suggest that these factors alter the validation process relative to PSG. However, the narrow participant group limits applicability to diverse populations. Future validation studies involving more ethnically and clinically diverse populations are warranted to enhance generalisability and to investigate these potential moderating effects.

Although we collected data from only 16 participants, each participant contributed up to four nights of recordings, resulting in 59 independent paired nights of data. We used linear mixed models to account for the repeated-measures design, thereby enhancing statistical power. A key strength of this study is the use of simultaneous PSG and actigraphy recordings over multiple nights, providing robust data for analysis. Furthermore, this study uniquely employs multiple statistical methods, including Bland–Altman plots, linear mixed models, and epoch-by-epoch comparisons, offering a comprehensive evaluation of the MotionWatch8’s performance. These statistical approaches are particularly well-suited to the experimental design and study population in this study, enhancing the validity of the findings. However, the controlled laboratory environment may limit generalisability, as natural sleep patterns could be influenced by fixed bedtimes. Additionally, MotionWatch8’s reliance on movement-based algorithms poses challenges in detecting wake epochs during periods of quiet rest, which remains a limitation of this device.

## 5. Conclusions

MotionWatch8 demonstrated overestimations for total sleep time (TST) and sleep efficiency (SE) and underestimations for sleep onset latency (SOL) and wake after sleep onset (WASO) compared to PSG. Bland–Altman plot analysis revealed significant proportional biases for SE, SOL, and WASO, while no proportional bias was observed for TST. Linear mixed model (LMM) analysis confirmed significant differences between the two devices for SOL, TST, and SE, and a marginally significant difference for WASO, indicating that MotionWatch8 and PSG do not produce interchangeable measurements for these parameters. Epoch-by-epoch comparisons revealed high sensitivity for sleep (94.8%) but low specificity (33.1%) for wake detection, underscoring the device’s challenges in accurately identifying wake epochs during periods of minimal movement. These findings suggest that while MotionWatch8 has potential for monitoring overall sleep patterns under warm laboratory conditions, it may be less reliable for individuals with more extreme sleep characteristics (e.g., insomnia), highlighting the need for caution in its use for detailed sleep assessments. Future studies should consider validating the MotionWatch8 in larger, more diverse populations, including real-world home settings and in individuals with diagnosed insomnia or thermoregulatory dysfunctions. Software improvements in scoring algorithms or device-specific calibration targeting extreme sleep characteristics may enhance accuracy in theses populations.

These findings contribute to the limited literature on actigraphy validation in thermally and hormonally sensitive populations and offer practical insights into its limitations under real-world clinical conditions. 

## Figures and Tables

**Figure 1 sensors-25-03040-f001:**
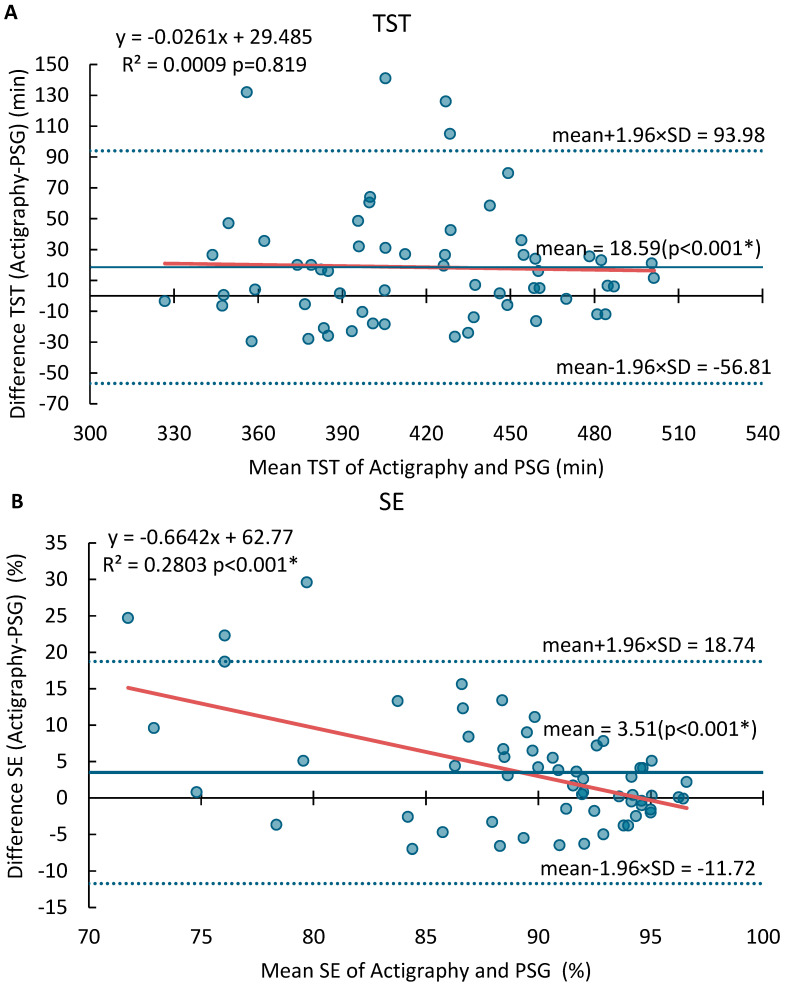
Bland–Altman analysis of agreement between MotionWatch8 and PSG for (**A**) Total sleep time (TST), (**B**) Sleep Efficiency (SE), (**C**) Sleep Onset Latency (SOL) and (**D**) Wake After Sleep Onset (WASO). Significant results (*p* < 0.05) are marked with *. The blue dots represent the differences between paired measurements (MotionWatch8—PSG) for each night, plotted against the mean of the two methods. The solid red line indicates the regression line showing proportional bias. The solid blue line indicates the mean differences. The dash blue lines indicate ±1.96 standard deviations.

**Table 1 sensors-25-03040-t001:** Participant characteristics.

Variable	n (%)	Mean± SD	Range
Age		51.4±4.2 years	45–63 years
BMI		26.0±3.1 kg/m^2^	20.7–30.0 kg/m^2^
**Menopause type**			
Natural	15 (93.8%)		
Surgical	1 (6.3%)		
**Menopause stage**			
Last menses < 1 year	8 (50.0%)		
1 year < last menses < 2 years	6 (37.5%)		
last menses > 2 years	2 (12.5%)		
**Dominant Hand**			
Right	13 (81.3%)		
Left	3 (18.8%)		
**Ethnicity**			
Asian	4 (25.0%)		
Caucasian	12 (75.0%)		

**Table 2 sensors-25-03040-t002:** Result of linear mixed model analysis.

Sleep Parameter	Device (Mean ± Std. Error)		F-Value	Devices
	PSG	MotionWatch8		
SOL (minutes)	16.0 ± 3.2	5.1 ± 3.2	9.835	0.004 *
WASO (minutes)	45.9 ± 6.6	36.2 ± 6.6	3.729	0.063 *
TST (minutes)	408.1 ± 11.0	426.8 ± 11.0	11.677	0.002 *
SE (%)	87.3 ± 1.6	90.9 ± 1.6	10.458	0.003 *

Significant results (*p* < 0.05) are marked with *.

**Table 3 sensors-25-03040-t003:** Sensitivity, specificity, and accuracy of MotionWatch8 compared to PSG.

	Mean ± SD	Range	95% CI
Sensitivity	94.8 ± 3.2%	85.8–100%	94.0%, 95.6%
Specificity	33.1 ± 19.5%	0.0–82.2%	28.1%, 38.1%
Accuracy	87.3 ± 6.6%	65.8–96.8%	85.6%, 88.9%

## Data Availability

Data are contained within the article.

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
