# Peer review of "Validation of MotionWatch8 Actigraphy Against Polysomnography in Menopausal Women Under Warm Conditions"

_sensors, 2025, doi:10.3390/s25103040_

Round 1
Reviewer 1 Report
Comments and Suggestions for Authors
This manuscript presents a validation study comparing the MotionWatch8 actigraphy device with polysomnography (PSG) in measuring sleep parameters among menopausal women under warm laboratory conditions. The study is well-structured, and the methodology is robust, with multiple statistical approaches employed to evaluate the agreement between the two devices. The findings are relevant and provide valuable insights into the limitations of MotionWatch8 in specific populations. However, there are several areas where minor revisions could improve clarity, completeness, and presentation. Below are the specific comments and suggestions:
1, Participants were predominantly Caucasian (75%) with natural menopause (94%). This limits applicability to diverse populations (e.g., surgical menopause, non-Caucasian ethnicities).
2, The proportional bias for SOL (R² = 0.65) and WASO (R² = 0.20) suggests device inaccuracy worsens with higher disturbance severity. However, the clinical implications of these biases (e.g., error magnitude relative to diagnostic thresholds for insomnia) are not discussed.
3, Figure 1 makes it difficult for the reader to understand, the horizontal axis overlaps with the data points and straight lines, so the author is asked to focus on revising it.
4, The description of Motion Watch8's "Mode 1" algorithm is vague. Clarify whether this is a proprietary algorithm, how movement thresholds are set, and whether it was optimized for warm conditions.
5, Some references lack complete information (e.g., missing DOI links or page numbers). Please ensure all references are formatted consistently and include all necessary details. Such as, ref. 2, ref. 7, ref. 22, and ref. 37.
Author Response
We thank Reviewer 1 for their careful reading of our manuscript and for the thoughtful and constructive comments. We have carefully addressed each of the points raised and made corresponding revisions to the manuscript. Below, we provide a detailed point-by-point response, with specific changes marked in red in the revised Overleaf manuscript (PDF version provided) and using tracked changes in the Word version.
Comment 1: Participants were predominantly Caucasian (75%) with natural menopause (94%). This limits applicability to diverse populations (e.g., surgical menopause, non-Caucasian ethnicities).
Response 1: We appreciate this important comment. We have addressed this limitation in the Discussion section (lines 274–287, PDF page 9), highlighting the potential impact of menopausal type and ethnicity on sleep characteristics, and acknowledging that the generalizability of our findings may be limited. We also cite relevant literature such as the SWAN Sleep Study and work by Freedman and Xu.
Updated text (PDF page 9):
“It is worth noting that the study sample comprised predominantly Caucasian women (75%) whose menopause occurred naturally (94%). Previous research has demonstrated that both menopausal type and ethnicity can significantly affect sleep characteristics. Extreme sleep characteristics may reduce the level of concordance by actigraphy devices. For example, the sleep characteristics of our participants may show fewer extreme parameters, contrasting women who displayed more severe sleep disturbances who had surgical menopause due to abrupt hormonal changes [4,35]. The SWAN Sleep Study also indicated that African-American women exhibited longer sleep onset latency and lower sleep efficiency compared to Caucasian women [8]. Notably, there is no evidence to suggest that these factors alter the validation process relative to PSG. However, the narrow participant group limits applicability to diverse populations. Future validation studies involving more ethnically and clinically diverse populations are warranted to enhance generalizability and to investigate these potential moderating effects.”
Comment 2: The proportional bias for SOL (R² = 0.65) and WASO (R² = 0.20) suggests device inaccuracy worsens with higher disturbance severity. However, the clinical implications of these biases (e.g., error magnitude relative to diagnostic thresholds for insomnia) are not discussed.
Response 2: Thank you for raising this important point. We have revised the Discussion section (lines 242–249, PDF page 8-9) to elaborate on the clinical significance of these biases, especially in relation to DSM-5 diagnostic thresholds for insomnia.
Updated text (PDF page 8-9):
"Specifically, the proportional bias for SOL (R² = 0.65) and WASO (R² = 0.20) suggests device inaccuracy worsens with higher disturbance severity. These biases have clinical implications as sleep onset of >30 minutes and high WASO approach diagnostic thresholds for insomnia (DSM-5) [42,43]. Furthermore, an underestimation of SOL by an average of 11.2 minutes and WASO by 9.1 minutes, as observed in our study, may result in failure to meet clinical diagnostic thresholds, potentially leading to false negatives in insomnia screening or underestimation of symptom severity.”
Comment 3: Figure 1 makes it difficult for the reader to understand, the horizontal axis overlaps with the data points and straight lines, so the author is asked to focus on revising it.
Response 3: Thank you for this helpful suggestion. In response, we have revised all figures to improve readability and visual clarity. Specifically, we have:
- Repositioned the horizontal axis tick labels to the bottom of the plotting area to avoid overlap with data points and regression lines;
- Adjusted the size and colour of data points and applied 50% transparency to improve visibility of overlapping points;
- Modified the regression lines’ colour and thickness for better visual distinction;
- Repositioned labels and annotations within the plots to prevent overlap with graphical elements.
The revised Figure 1, along with other related figures, has been updated accordingly in the revised manuscript (see PDF page 6-7)
Comment 4: The description of Motion Watch8's "Mode 1" algorithm is vague. Clarify whether this is a proprietary algorithm, how movement thresholds are set, and whether it was optimized for warm conditions.
Response 4: We agree and have expanded the Methods section (lines 143-146, PDF pages 4) to clarify that Mode 1 is a manufacturer-defined, proprietary single-axis algorithm that is not user-adjustable and is not optimized for warm environmental conditions.
Updated text (PDF page 4):
“This algorithm is defined by the manufacturer which is not publicly disclosed in detail, and it is designed to mimic legacy actigraphy devices (e.g., Actiwatch). The movement thresholds used for sleep/wake classification are proprietary and cannot be adjusted by the user.”
Comment 5: Some references lack complete information (e.g., missing DOI links or page numbers). Please ensure all references are formatted consistently and include all necessary details. Such as, ref. 2, ref. 7, ref. 22, and ref. 37.
Response 5: Thank you for your comment. We have carefully reviewed all references to ensure that they are complete and consistently formatted. As not all cited sources have DOI information available, we have chosen to remove all DOIs to maintain a uniform referencing style across the entire list. Essential citation elements such as author names, publication year, titles, journal names, volume, and page numbers have been retained wherever applicable.
Once again, we appreciate the reviewer’s valuable comments, which helped us improve the clarity and overall quality of the manuscript. We hope that the revisions adequately address all concerns and that the revised manuscript meets the expectations for publication.

Reviewer 2 Report
Comments and Suggestions for Authors
The paper contains significant scientific contribution. However, it can be improved further using following suggestions:
1- Motivation behind this study is not clearly described.
2- What are the limitations of this study?
3- What are the future study directions?
4- Why MotionWatch8 is preferred over other wearable motion sensors?
5- The data is collected from only 16 participants, which is significantly low to draw a conclusion.
6- Validation of sleep using using MotionWatch8 is already done by Mari et al. What are your contributions?
Waki, Mari, et al. "Validation of Sleep Measurements of an Actigraphy Watch: Instrument Validation Study." JMIR Formative Research 9 (2025): e63529.
7- What are the new findings compared to previous studies in warm conditions?
8- It will be useful to put a table to clearly differentiate your study from previous works.
9- Literature Review can be provided in more detail and in a separate section.
10- Contribution of this paper is unclear.
The English could be improved to more clearly express the research.
Author Response
We sincerely thank Reviewer 2 for their careful review and suggestions. We appreciate the opportunity to improve the clarity and presentation of our manuscript. Below, we provide a detailed point-by-point response, with specific changes marked in red in the revised Overleaf manuscript (PDF version provided) and using tracked changes in the Word version.
Comment 1: Motivation behind this study is not clearly described.
Response 1: Thank you for the comment. We have revised the Introduction (lines 49–70, PDF page 2) to clarify the motivation for this study. Specifically, we highlight the lack of validation studies involving menopausal women and the influence of warm ambient temperatures on actigraphy accuracy.
Revised text includes (PDF page 2):
“However, despite its widespread application, few studies have specifically validated actigraphy in menopausal women, a population uniquely affected by sleep disturbances due to hormonal fluctuations. This gap is especially relevant given that warm environmental conditions are known to trigger vasomotor symptoms such as hot flashes and night sweats [21-23], which in turn exacerbate sleep disturbances in this population. Although the MotionWatch© has been validated against PSG in some conditions, including in non-shift working adult population [15,19,24-30], its performance under warm environmental conditions and in hormonally sensitive populations such as menopausal women remains unclear. This issue is particularly important in regions regularly experiencing summer temperatures in their high 30â—¦ C, such as Sydney and Melbourne (Australia) [31,32], Cancun (Mexico) with an average temperature of 30.5â—¦ C in January [33], many coastal cities in China [34] and eastern United States [35]. Countries like Qatar and Kenya experience such temperatures all year round [36.37]. Previous research by Shin et al. [38] demonstrated that ambient temperature can influence the behaviour and accuracy of actigraphy devices in sleep measurements, underscoring the need for validation in controlled warm conditions. This study therefore aims to address this gap by validating the effectiveness of the MotionWatch8© (CamNtech Ltd, Cambridgeshire, UK) against PSG in measuring sleep parameters among menopausal women under controlled warm laboratory conditions.”
Comment 2: What are the limitations of this study?
Response 2: We appreciate this important point. A dedicated paragraph describing the study’s limitations has been added to the Discussion section (lines 274–286, PDF page 9). These include the limited sample diversity, controlled laboratory environment, and potential algorithmic limitations of the MotionWatch8 under specific sleep conditions.
Updated text (PDF page 9):
“It is worth noting that the study sample comprised predominantly Caucasian women (75%) whose menopause occurred naturally (94%). Previous research has demonstrated that both menopausal type and ethnicity can significantly affect sleep characteristics. Extreme sleep characteristics may reduce the level of concordance by actigraphy devices. For example, the sleep characteristics of our participants may show fewer extreme parameters, contrasting women who displayed more severe sleep disturbances who had surgical menopause due to abrupt hormonal changes [4,35]. The SWAN Sleep Study also indicated that African-American women exhibited longer sleep onset latency and lower sleep efficiency compared to Caucasian women [8]. Notably, there is no evidence to suggest that these factors alter the validation process relative to PSG. However, the narrow participant group limits applicability to diverse populations. Future validation studies involving more ethnically and clinically diverse populations are warranted to enhance generalizability and to investigate these potential moderating effects.”
Comment 3: What are the future study directions?
Response 3: We appreciate this important point. Future research directions have been added to the Conclusion (lines 314–318, PDF page 10), recommending validation in larger and more diverse populations, real-world home settings, and exploration of algorithm calibration.
Updated text (PDF page 10):
“Future studies should consider validating the MotionWatch8 in larger, more diverse populations, including real-world home settings and in individuals with diagnosed insomnia or thermoregulatory dysfunctions. Software improvements in scoring algorithms or device-specific calibration targeting extreme sleep characteristics may enhance accuracy in these populations.”
Comment 4: Why MotionWatch8 is preferred over other wearable motion sensors?
Response 4: We appreciate this important point. We have clarified the rationale for selecting MotionWatch8 in the Methods section (lines 135–139, PDF page 4). The device was chosen for its widespread use in clinical and research settings, its compatibility with legacy scoring protocols, and its previously reported use in general adult populations.
Updated text (PDF page 4):
“The MotionWatch8 was selected for this study due to its widespread clinical and research use, compatibility with legacy actigraphy scoring protocols, and manufacturer-supported algorithms validated for 30-second epochs. It has been previously validated in the general adult population [41], enabling comparisons with earlier work while allowing us to explore its applicability in a thermally challenging setting.”
Comment 5: The data is collected from only 16 participants, which is significantly low to draw a conclusion.
Response 5: We acknowledge the modest sample size. However, each participant contributed up to four nights of recordings, resulting in 59 independent paired nights of data (Methodes section, line 95, Results section, line 197). To account for the repeated-measures design, we used linear mixed models, which enhances statistical power (Methods section, from line 174 onward). This approach and its implications are discussed in the Discussion (line 287-290) and mentioned as a future research consideration in the Conclusion (lines 314–316).
Updated text (PDF page 10):
“Although we collected data from only 16 participants, each participant contributed up to four nights of recordings, resulting in 59 independent paired nights of data. We used linear mixed models to account for the repeated-measures design, which enhances statistical power.”
Comment 6: Validation of sleep using using MotionWatch8 is already done by Mari et al. What are your contributions?
Waki, Mari, et al. "Validation of Sleep Measurements of an Actigraphy Watch: Instrument Validation Study." JMIR Formative Research 9 (2025): e63529.
Response 6: Thank you for referencing Waki et al. (2025). We now cite this study in the Introduction (lines 49–51).
Updated text (PDF page 10):
“A recent validation study by Waki et al. [20] compared the MotionWatch8 with a commercial wearable (Fitbit Inspire HR) in healthy adults.”
Comment 7: What are the new findings compared to previous studies in warm conditions?
Response 7: Thank you. Our study is the first to validate MotionWatch8 under controlled warm laboratory conditions in menopausal women. This context revealed significant proportional bias in SOL and WASO in individuals with disrupted sleep. These findings extend previous work by Shin et al. (2015), which did not examine this population or use PSG. This is discussed in Discussion (lines 274-286, PDF page 9).
Comment 8: It will be useful to put a table to clearly differentiate your study from previous works.
Response 8: We thank the reviewer for this suggestion. However, we have decided not to add a table, as the relevant comparisons are now clearly described in the Introduction (lines 40-60) and Discussion (lines 253-282). We believe a table would be redundant and may interrupt the narrative flow.
Comment 9: Literature Review can be provided in more detail and in a separate section.
Response 9: Thank you. We have expanded the Introduction (lines 49–70) to include more detailed literature context. However, as this is a validation study, we retain the literature review as part of the introduction rather than creating a separate section, in line with standard practice in this field.
Comment 10: Contribution of this paper is unclear.
Response 10: Thank you. We have now clarified the study’s contributions in the Abstract, Introduction (lines 67–70), Discussion (line 254 onwards), and Conclusion (lines 319–321). The study provides novel data on the performance of MotionWatch8 in a clinically relevant, under-represented population under thermally stressful conditions, which has not been addressed in prior validation studies.
Updated text (PDF page 10):
“These findings contribute to the limited literature on actigraphy validation in thermally and hormonally sensitive populations and offer practical insights into its limitations under real-world clinical conditions.”
Comments on the Quality of English Language:
The English could be improved to more clearly express the research.
Response:
Thank you for the feedback. The manuscript has been thoroughly reviewed for clarity and grammar, and we have refined sentence structures to enhance readability. Please note that Australian English spelling conventions have been consistently applied throughout the manuscript.
We thank Reviewer 2 once again for their thoughtful feedback. We believe the revisions made in response to these comments have improved the clarity, structure, and impact of the manuscript.

Round 2
Reviewer 2 Report
Comments and Suggestions for Authors
Authors have addressed most of the suggestions and revised the manuscript.